

# Experimental food supplementation increases reproductive effort in the Variable Antshrike in subtropical Brazil

James J. Roper[1,2], André M.X. Lima[1] and Angélica M.K. Uejima[2,3]

[1] Programa de Pós Graduação em Ecologia e Conservação, Universidade Federal do Paraná, Curitiba, Paraná, Brazil
[2] Departamento de Zoologia, Universidade Federal do Paraná, Curitiba, Paraná, Brazil
[3] Centro Acadêmico de Vitória, Universidade Federal de Pernambuco, Vitória de Santo Antão, Pernambuco, Brazil

## ABSTRACT

Food limitation may interact with nest predation and influence nesting patterns, such as breeding season length and renesting intervals. If so, reproductive effort should change with food availability. Thus, when food is limited, birds should have fewer attempts and shorter seasons than when food is not limiting. Here we experimentally test that increased food availability results in increased reproductive effort in a fragmented landscape in the Variable Antshrike (*Thamnophilus caerulescens*) in southern Brazil. We followed nesting pairs in a naturally fragmented habitat and experimentally supplemented food for half of those pairs. Birds were seen, but evidence of nesting was never found in two small fragments, even though these fragments were larger than individual territories. Pairs with supplemented food were more likely to increase clutch size from two to three eggs and tended to renest sooner (20 d on average) than control pairs. Also, fragment size was associated with breeding patterns, although fragment replicates were unavailable. Nest duration, nest success and breeding season length were all greater, while renesting intervals were shorter, in the largest fragments. Simulations showed that only the largest fragments were able to have a net production of young. Food availability clearly influenced reproductive effort and as a consequence, because of the interaction with predation risk, forest fragments of varying sizes will have complex reproductive dynamics.

## INTRODUCTION

Nest predation is the greatest cause of nesting failure among open nesting passerine birds and is likely to have influenced avian life-history evolution (*Nice, 1957*; *Skutch, 1949*; *Skutch, 1985*; *Ricklefs, 1969*; *Ricklefs, 2000a*; *Ricklefs, 2000b*; *Roper, Sullivan & Ricklefs, 2010*). Food is also important and can limit reproduction in birds (*Ricklefs, 1968*; *Ricklefs, 2010*; *Martin, 1987*; *Derbyshire, Strickland & Norris, 2015*) and may interact with predation, thereby causing complex (behavioral and life history) responses to predation risk. For example, the seasonal decline in clutch size in North American passerine birds may be due to

Corresponding author
James J. Roper, jjroper@gmail.com

reduced food availability with each nesting attempt (*Martin, 1987*; *Milonoff, 1991*; *Bauchau & Seinen, 1997*; *Castro et al., 2003*). Where breeding seasons are long, nest predation rates are high and birds are "income breeders" (that is, food availability at the time of egg production limits egg production, rather than stored reserves that fuel egg production in "capital breeders"), food availability may limit annual reproductive success because of its influence on both the number of nesting attempts and individual nest success (*Soler & Soler, 1996*; *Davis, Nager & Furness, 2005*; *Ricklefs, 2010*; *Roper, 2005*; *Stephens et al., 2014*). Lower food abundance can result in fewer renesting attempts following predation (*Rolland, Danchin & Defraipont, 1998*; *Roper, Sullivan & Ricklefs, 2010*; *Zanette et al., 2011*) and more food may increase nesting success since both young and parents may be well-fed by fewer trips to the nest, thereby reducing the potential effect of visitation rate on predation risk (*Holmes et al., 1992*; *Kuituken & Makinen, 1993*; *Soler & Soler, 1996*; *Martin et al., 2011*). Thus, experimentally increased food abundance may reduce the incubation period (if it is flexible and not genetically constrained), and reduce nest predation (because of fewer trips to and from the nest), increase growth rates and permit additional nesting attempts in species that usually have only one successful nest per year (*Martin, 1987*; *Davis & Graham, 1991*; *Meijer & Drent, 1999*; *Castro et al., 2003*; *Roper, Sullivan & Ricklefs, 2010*).

Food abundance may interact synergistically with fragment size, predation risk and other causes of nest failure (*Gates & Gysel, 1978*; *Donovan et al., 1995*; *Donovan et al., 1997*; *Burke & Nol, 1998*; *Batáry & Báldi, 2004*; *Huhta, Jokimäki & Rahko, 1998*; *Huhta et al., 2004*; *Abensperg-Traun, Smith & Main, 2000*, but see *Tewksbury, Hejl & Martin, 1998*). However, even though both predation risk and food abundance may be independently influenced by fragment size (*Askins, 1995*; *Melo & Marini, 1997*; *Weinberg & Roth, 1998*; *Stratford & Stouffer, 2001*; *Fort & Otter, 2004*; *Tewksbury et al., 2006*), declining food abundance may still influence the likelihood of repeated nesting attempts following predation due to a reproductive cost or decreased survival (*Ruiz-Gutiérrez, Gavin & Dhondt, 2008*). Thus, annual reproductive success is expected to decline due to an inverse relationship between both predation risk and food abundance with fragment size. When predation rates are high renesting is even more important for annual reproductive success (*Roper, 2005*; *Roper, Sullivan & Ricklefs, 2010*). Therefore, a consequence of high predation rates, renesting rates may decline and brood reduction may occur more often in smaller fragments due to reduced food abundance (*Suarez, Pfenning & Robinson, 1997*; *Huhta, Jokimäki & Rahko, 1998*; *Zanette & Jenkins, 2000*; *Barding & Nelson, 2008*; *Hinam & St Clair, 2008*).

The landscape of fear concept (*Bleicher, 2017*; *Creel, 2018*) may be applicable in fragmented landscapes if the probability of nest predation is associated with fragment size. Thus, we may expect complex interactions and responses to both food abundance and nest predation in birds that are found in fragmented landscapes (*Zanette et al., 2011*). How those interactions are manifest in nature remain to be studied because of their complexity.

Here, we examine nesting success and experimentally manipulated food abundance in a fragmented landscape to test for the importance of food abundance and predation risk and their interactions in a subtropical understory-nesting bird, the Variable Antshrike (*Thamnophilus caerulescens*, Vieillot, 1816) in southern Brazil. The Variable Antshrike was chosen because the genus *Thamnophilus* has been studied in the tropics and so its breeding

biology may be compared with other species in the genus (*Roper, 2003*; *Roper, 2005*; *Roper & Goldstein, 1997*; *Tarwater, 2008*; *Roper, Sullivan & Ricklefs, 2010*; *Tarwater & Brawn, 2010*). Also, it is a relatively common, yet poorly studied species of the Atlantic Forest southeastern South America (*Oniki & Willis, 1999*). Finally, visits to the study area always found antshrikes in the visited forest fragments. We predict that food supplementation will result in increased reproductive effort, which may be manifest as increased number of nesting attempts, reduced intervals between nesting attempts, greater nesting success and combinations thereof (*Ruffino et al., 2014*).

## METHODS

### Study area

Nesting in the Variable Antshrike was studied in a region of natural forest fragments separated by open grassland and savanna, Vila Velha State Park (25.25°S, 50.08°W, ~1,000 m above sea level) in Paraná, southern Brazil (Figs. 1 and 2). The entire park comprises 3,122 ha and the forest fragments are typical of the mixed Atlantic Forests of Brazil, dominated by *Araucaria angustifolia*. Natural fragments range in size from 1 to 450 ha and are separated from each other by a minimum of 50 m. We selected five fragments (with >100 m separation from each other, and from other fragments) with areas of 4, 23, 24, 112 and 214 ha to include the approximate range of local fragment size and maintain the >100 m distance between fragments. Due to time constraints, it was not feasible to include more fragments in this study, hence we will emphasize the analysis of food supplementation.

### Study species

Nesting in the Variable Antshrike *Thamnophilus caerulescens* (Vieillot, 1816) (Passeriformes: Family Thamnophilidae) was monitored in the 2000 to 2002 breeding seasons that begin in October and end in January of the subsequent year. The antshrike is a sexually dimorphic, insectivorous, understory bird. As in other members of the family, it is territorial, monogamous (as far as is known), and nests in the forest understory, building an open-cup nest in horizontal forks of shrubs and saplings usually less than 3 m above the ground, as is typical of the genus (*Oniki, 1975*; *Ridgely & Tudor, 1994*; *Roper, 2000*; *Roper, 2005*). Variable Antshrikes lay clutches of two or three eggs (see below) and both parents contribute in all aspects of reproduction, including nest construction, incubation, feeding nestlings and post-fledging care (*Oniki & Willis, 1999*; *Zimmer & Isler, 2003*). Details of the breeding season, such as start and end of the breeding season, number of clutches per year, variation in clutch size, are all very poorly know, and will be reported here for the first time.

We captured Variable Antshrikes using mist-nets and banded each with a unique combination of colored leg bands and a metal band provided by CEMAVE (the Brazilian governmental agency that oversees bird studies). Because we wished to know annual reproductive success, we followed banded pairs to find nests (each year attempting to capture more pairs). Song playbacks to attract birds to the nets were sparingly used when pairs were difficult to find. Nests were checked every 2–3 days until they either succeeded
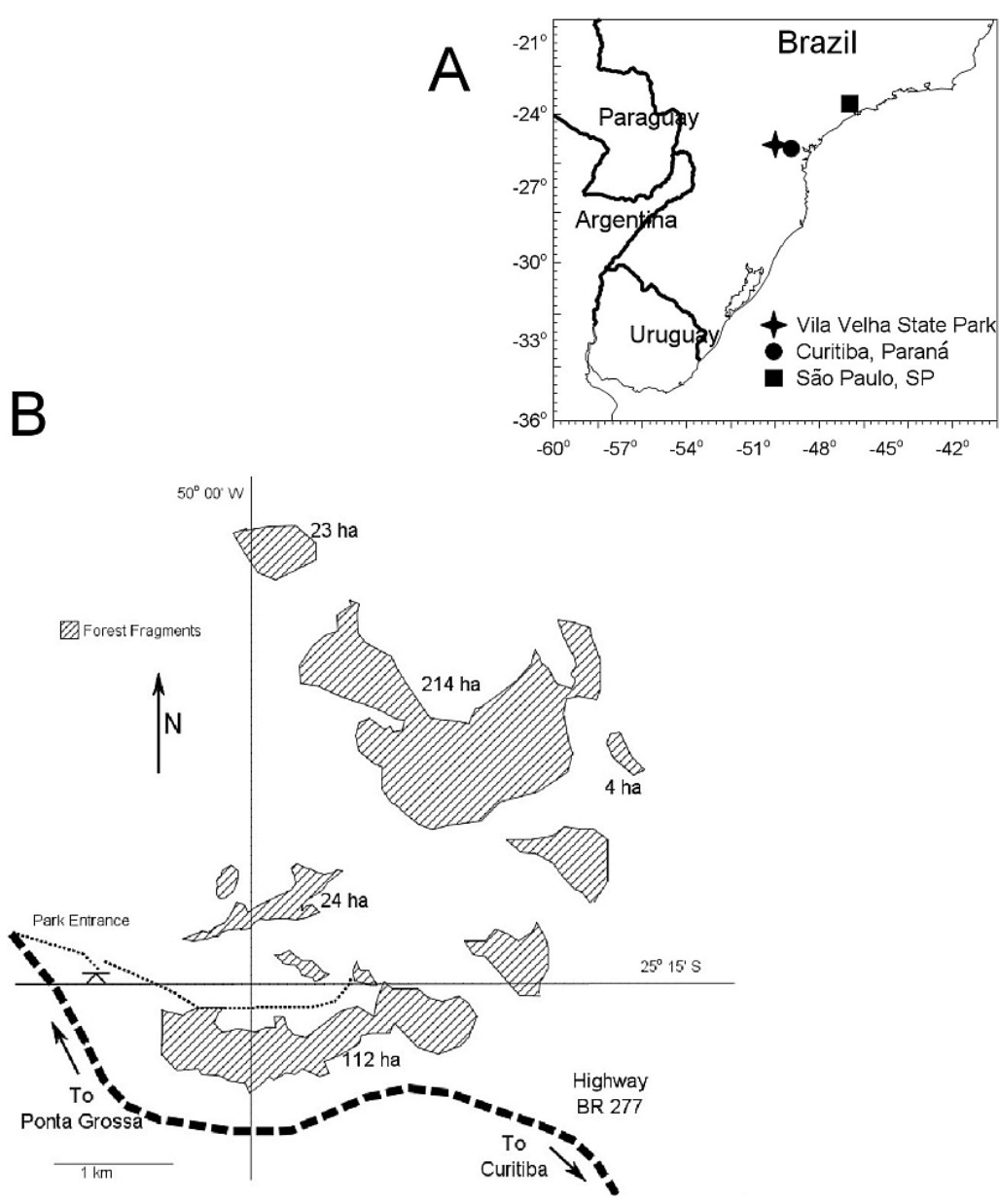

**Figure 1** **Study area location and configuration of forest fragments.** (A) Location of Vilha Velha State Park where this study took place, in southern Brazil. (B) Schematic map of the study area indicating the forest fragments included in this study.

(when one or more young fledged) or failed (when no young fledged). Predation was assumed to be the cause of failure when eggs or nestlings disappeared prior to the fledging date and adults were not seen with young birds away from the nests. All methods and animal manipulation followed standards of ethics under Brazilian law.

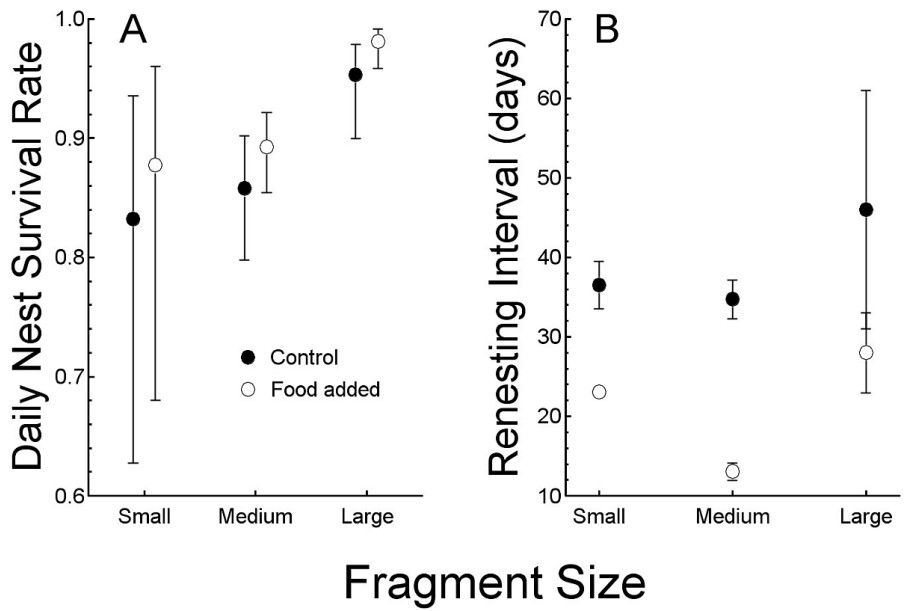

**Figure 2 Comparisons of the response to food treatments (control and added food).** Comparisons of the response to food treatments (control and added food) of (A) daily nest survival rates (mean, with 95% confidence interval) and (B) renesting intervals in days. Both are compared by forest fragment sizes to illustrate interactions.

## Experimental food supplementation

To test the importance of food on nesting success about half of the pairs found in each fragment were randomly chosen to receive supplemental food. In these territories, food (mealworms, Tenebrionidae) was placed in a small dish (∼10 cm in diameter, used for feeding cage birds, about the same diameter as a nest; (*Zimmer & Isler, 2003*; *Londoño, Levey & Robinson, 2008*) attached to a branch within 5 m of the first active nest found for each pair. We did not attempt to quantify how much food we gave, but placed ∼10 mealworms in each dish each day, until fledging or nest failure. If a nest failed and when a new nest of that same pair was found, the dish was moved to about 5 m from the new nest and food was again made available. While the birds were accustomed to feeding from the dish where it was, we moved it closer to the new nest for both logistical reasons and to help ensure that the same pair fed from the dish. We visually confirmed that all pairs with supplemented food took the offered mealworm larvae. Assignment of experimental treatments were constant (each pair was used for only one treatment) throughout the study. Because food supplementation began with the first nest attempt found for pairs in the supplementation treatments, food supplementation was not important in first nest initiation (see below). Also, because the food dish was small and inconspicuous, we assumed that the presence of a dish itself was unimportant, therefore we did not put empty dishes in the control territories.

## Reproductive effort

Effort was measured in four main ways: the number of nesting attempts, length of the renesting interval, breeding season length and nesting success. Because our study area is fragmented, we also tested for interactions with fragment size (area) as a covariate with interaction terms when sample sizes allowed. Four potential outcomes of the experiment were possible with respect to effort: (1) effort is independent of food treatment and fragment size, (2) effort increases with supplemented food but is independent of fragment size, (3) effort is independent of food treatment but dependent on fragment size and (4) effort increases with both supplemented food and fragment size.

We tested these possibilities with nesting success using the program MARK (version 6.0). MARK permits the estimation of daily survival rate that may be compared (using ΔAICc to select the best model) among groups (*White & Burnham, 1999*; *Dinsmore, White & Knopf, 2002*). We compared renesting intervals, number of renesting attempts, among food treatments and fragments of different size and their possible interactions using analysis of variance (ANOVA) or analysis of covariance (ANCOVA) when appropriate. Only the intervals following failed nesting attempts were used in this analysis because the interval after success involves an extended period of post-fledging care. We compared clutch size among treatments (both fragment size and food supplementation) using log-likelihood ratio test (*G*).

Finally, while we predicted that annual productivity should reflect effort, in practice, annual productivity is a complex interaction between predation rate and number of nesting attempts (*Roper, Sullivan & Ricklefs, 2010*). Once a nest is successful, parents must then spend some amount of time in parental care of fledglings. Also, if the number of successes are limited, even in relatively poor conditions, by renesting pairs may finally beat the odds and successfully nest (*Roper, 2005*). Thus, to examine how annual productivity might change with food abundance and the correlated variables (renesting intervals, breeding season length) we simulated the annual breeding cycle (see *Roper, Sullivan & Ricklefs, 2010*) using the parameters found in this study (daily nest survival rates, renesting intervals and clutch size by fragment size), and generated 100 replicates of 30 pairs under each combination of conditions. We compared annual productivity under the combinations of conditions using multivariate analysis of variance (MANOVA). Under the assumption that the parameters measured and estimated in this study reflect actual field conditions, this simulation allows generating larger samples that are then comparable (*Roper, Sullivan & Ricklefs, 2010*).

## RESULTS

During preliminary censuses prior to beginning this experiment, we found antshrikes in all fragments. However, while capturing and marking birds, we never captured antshrikes nor found evidence of nesting in the smallest fragment (4 ha), even though observations suggest that their territory size can be much smaller (~1 ha, J Roper, pers. obs., 2002–2018). We state this for two reasons: (1) we attempted replicates of small fragments, (2) to demonstrate that small fragments may be unsuitable, perhaps for reasons we will describe here. Nine

females were captured in the larger of the two small fragments (23 ha), but never again seen again in the fragment in which they were captured. Four of them were later found and followed in the 112 ha fragment where they remained in the same territory for the duration of this study. Nests (a total of 103) were only found in use in the 24, 112 and 214 ha fragments, and so food supplementation was only possible in those fragments. A total of three pairs were followed in the 24 ha fragment, and eight pairs each in the 112 and 214 ha fragments. Once pairs were marked and followed, most pairs remained together for the duration of this study. When an individual disappeared, it was replaced by a previously unbanded bird and no "divorce" or mate-switching was ever observed.

## Timing of Breeding

We describe the second two years because during the first year we spent time early in the season capturing birds and may have missed some nests. During the next two years of this study the breeding season began on 26 Oct 2001 (first egg laid the following day) and 9 Oct 2002 (with the first of two eggs) and ended on 26 Jan 2002 (last nest found with 2 eggs) 9 Jan 2003 (found with two eggs, 92 days). Breeding season length varied by fragment size. Pairs in the 112 ha fragment had the longest season length (91 d in 2001, 89 d in 2002), followed by pairs in the 214 ha (77 d in 2001, 82 d in 2002) and 24 ha (42 d in 2001, 48 d in 2002).

## Reproductive effort and success

Nesting success (as daily nest survival rate) was lowest in the 11 nests (all of which failed) in 2000 ($0.836$ day$^{-1}$, SE $= 0.045$). We do not include those nests in analysis because it being the first year of study, when we were also actively capturing birds, we cannot be certain that our activities did not affect success, and sample size was small. In subsequent years, the field season began with banded birds, so we captured less often, thereby reducing our potential influence on nesting. Thus, in 2001 ($0.911$ day$^{-1}$, SE $= 0.014$, $N = 41$) and 2002 ($0.935$ day$^{-1}$, SE $= 0.011$, $N = 51$), daily survival rates were similar. None of the seven nests was successful in the small (24 ha) forest fragment. Five of 67 nests (8%) were successful in the 112 ha fragment and 17 of 29 nests (59%) were successful in the 214 ha fragment (Table 1). Daily survival rate (DSR) was consequently greatest in the large fragment ($0.97$ day$^{-1}$, SE $= 0.01$), which was greater than that of the other two fragments (in which DSR was similar: 112 ha $= 0.88$ day$^{-1}$, SE $= 0.01$, 24 ha $= 0.85$ day$^{-1}$, SE $= 0.05$, $P < 0.05$). The large differences in daily survival rate among the fragments dominates the relationship between food abundance and nesting success and the model with only forest fragment size as a predictor variable and the model including supplemented food and fragment size were similar (other models had $\Delta$AICc $>10$, likelihood ratio test, $\chi^2 = 5.0$, SE $= 3.0$, $P > 0.05$, Table 2, Fig. 2A).

The combined effect of food supplementation and forest fragment size (with no interaction) explained 37% of the variance in number of nesting attempts per pair per year, with 26% explained by supplementation and 11% by fragment size ($F_{3,35} = 8.58$, $R^2 = 0.37$, $P < 0.001$). With supplemented food, the number of nesting attempts increased to an average of 2.17 nests year$^{-1}$ (SE $= 0.47$) as compared to 1.3 nests (SE $= 0.13$) in

**Table 1 Comparison among the reproductive parameters, forest fragment size and supplemental food treatment.**

| Fragmentu size (ha) | Food treatment | Pairs (N) | Nesting attempts (N) | Successes (N) | Nest DSR[a] (SE) | Renesting interval days[b] |
|---|---|---|---|---|---|---|
| 24 | Control | 1 | 4 | 0 | 0.85 | 35–38 (3) |
| 24 | Fed | 2 | 3 | 0 | (0.05) | 23 (1) |
| 112 | Control | 4 | 27 | 2 | 0.88 | 25–44 (26) |
| 112 | Fed | 4 | 40 | 3 | (0.02) | 3–18 (40) |
| 214 | Control | 3 | 10 | 4 | 0.97 | 23–54 (9) |
| 214 | Fed | 5 | 19 | 13 | (0.01) | 14–45 (3) |

Notes.
[a] DSR was similar for control and fed treatments, but was greater in the large fragments ($P < 0.05$).
[b] Minimum and maximum intervals in days, number of nests in parentheses is different from the total number of nesting attempts because only intervals after failed nests are included.

**Table 2 Model comparison of the interactions between nesting success, fragment size and experimental food supplementation (Fig. 3).** Fragment size had the lowest AICc value at 415.4.

| Model | ΔAICc | Parameters |
|---|---|---|
| Fragment size | 0 | 3 |
| Fragment size + Food | 2.2 | 6 |
| Food | 27.4 | 2 |
| Constant | 30.1 | 1 |

controls ($F_{3,38} = 8.5$, $r^2 = 0.37$, $p < 0.05$). The 112 ha fragment had the greatest number of attempts pair$^{-1}$ year$^{-1}$ (least squares mean = 3.2, SE = 0.2, maximum = 5) followed by the 214 ha fragment (1.95, SE = 0.24, max = 3) and 24 ha fragment (mean = 1.75, SE = 0.45, max = 2, $P < 0.05$).

Variable Antshrikes may renest after both failed and successful nests. In all years, pairs varied from no renesting attempts (three pairs, one both years), one renesting ($N = 16$), two ($N = 9$), three ($N = 5$), and four ($N = 4$), for a total of 65 renesting attempts. Overall, renesting occurred from 3 to 54 days after failure (median = 15 days) and 11–45 days after success (median = 26 days, Table 1). We do not have data that indicate whether the renesting after success was influenced by post-fledging mortality or dispersal.

Renesting interval was independent of year (ANOVA, $F_{2,52} < 1$, $P > 0.5$) and so we combined years in the following analysis. Renesting interval after unsuccessful nests was influenced by both, food supplementation and fragment size with no interaction (ANCOVA, adjusted $R^2 = 0.87$, $F_{3,47} = 110.7$, $P < 0.001$, Table 1, Fig. 2B). Of the variance explained by the full model (87%), 65% was due to the addition of food and the remaining 12% was explained by fragment size. In control territories, renesting interval was shorter by 7.8 d in the large (214 ha, 28 d) versus the small (24 ha, 35 d) fragment which was similar to the 112 ha (34 d) fragments. In food supplemented territories, renesting in the large fragment and medium fragment (15 and 12 d respectively) were similar, and less than that in the small fragment (22 days, Table 1).

Clutch size varied between two and three eggs, and 23 of the 25 three-egg clutches followed food supplementation. Only two-egg clutches were found in the smallest fragment
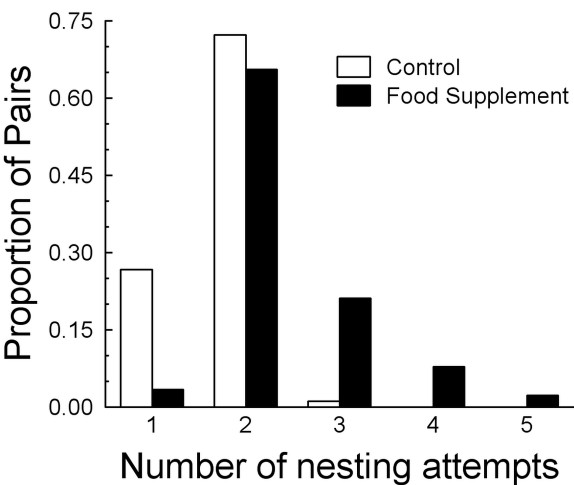

**Figure 3** **Comparison of number of nesting attempts per year between control and food treatments.** Comparison of the distributions of the number of nesting attempts expected based on simulations of the consequences of natural (Control) and supplemented food based on the nesting data in Table 3. Only birds with supplemented food can attempt more than three nests each breeding season.

($n = 7$ clutches), 2 ($n = 51$) and 3 ($n = 16$, 24%) egg clutches in the 112 ha fragment and 2 ($n = 20$) and 3 egg ($n = 9$, 31%) clutches in the 214 ha fragment. Due to limited sample size, we could not test an additional effect of fragment size on clutch size.

As predicted, the probability of renesting increased in food supplemented pairs, with an interaction with fragment size ($F_{5,174} = 29.0$, $R^2 = 0.44$, $P < 0.0001$). In the full model, 21% of the 44% of the variance was explained by food supplementation, 18% by fragment size, and the remaining 5% by the interaction. In the simulations, up to five attempts were possible (Fig. 3) and annual productivity was always greatest with supplemented food (Table 3, Fig. 4). Also, multiple successful nests within a year were only likely to occur in the food supplemented scenarios (Table 3), and the four pairs that had two successful nests in 2002 all received supplemented food. Failure to reproduce in a give breeding season was likely to occur in a large proportion of the control pairs and occurred in five (2001) and four (2002) control pairs, and three (both years) fed pairs.

## DISCUSSION

In this first experimental test of food supplementation as an influence of parental nesting effort in neotropical birds, three trends clearly demonstrate that food abundance can influence effort and, as a consequence, annual reproductive success. When food was added, (1) pairs renested after shorter intervals, (2) pairs increased their number of nesting attempts per year, and (3) females laid larger clutches. Further, preliminary evidence suggests that fragment size also matters even though an ideal fragmentation study should include replicates of fragments (unavailable in this study area). Breeding began sooner and ended later in larger fragments, nesting success was greatest in the largest fragment, all nests failed in the smallest fragment, and all nest failure was due to predation. Also,
**Table 3** Data-based simulation (100 runs of 30 simulated pairs) results with values used in the simulation (daily survival rate—DSR, clutch size—CS, breeding season length—BSL, renesting interval after failure and success), average number of nests pair$^{-1}$ (Attempts) to achieve the resulting Productivity (average number of offspring pair$^{-1}$), and the percentage of successful nests (Rate) and the probability of a second successful nest during the season (>1 success). See Fig. 4.

| Fragment (DSR) | Food | CS | BSL | Renest[a] | | Attempts | Productivity (SD) | Success rate | |
|---|---|---|---|---|---|---|---|---|---|
| | | | | Failure | Success | | | Rate | >1 success |
| Small 0.85 | Control | 2 | 45 | 37 | 48 | 1.816 | 0.074 (0.378) | 3.7 | |
| | | 3 | | 39 | 50 | 1.614 | 0.126 (0.602) | 4.2 | 0 |
| | Added | 2 | | 23 | 34 | 2.138 | 0.094 (0.423) | 4.7 | |
| | | 3 | | 25 | 37 | 2.088 | 0.150 (0.654) | 5.0 | |
| Medium 0.88 | Control | 2 | 90 | 34 | 45 | 2.837 | 0.348 (0.779) | 17.0 | 0.004 |
| | | 3 | | 36 | 47 | 2.754 | 0.414 (1.069) | 13.4 | 0.004 |
| | Added | 2 | | 14 | 25 | 4.732 | 0.516 (0.916) | 24.9 | 0.009 |
| | | 3 | | 16 | 27 | 4.400 | 0.642 (1.309) | 20.3 | 0.011 |
| Large 0.97 | Control | 2 | 80 | 53 | 64 | 1.516 | 1.458 (0.889) | 72.9 | 0 |
| | | 3 | | 55 | 66 | 1.484 | 2.244 (1.303) | 74.8 | 0 |
| | Added | 2 | | 26 | 37 | 2.195 | 2.144 (1.260) | 83.5 | 0.237 |
| | | 3 | | 28 | 39 | 2.162 | 3.114 (1.960) | 80.5 | 0.233 |

**Notes.**
[a] Failed nests are the most common so we calculated renesting interval after failure for each treatment. Renesting interval after success was the average of all observed renesting intervals plus 11 days. We added a cost to three-egg clutches as two additional days in the nesting cycle, because egg laying occurs every other day.

we emphasize that two smaller fragments (in which we previously found singing birds) had no nesting attempts and during this study, birds were not consistently found in these fragments, thus indicating that small fragments are problematic.

Food supplementation reduced renesting intervals and so with added food, more rapid renesting, and consequently more nesting attempts summed to greater annual success (*Roper, 2005*; *Roper, Sullivan & Ricklefs, 2010*; *Ruffino et al., 2014*). Food availability may also interact with other variables (self-maintenance, time incubating, etc.) during nesting (*Londoño, Levey & Robinson, 2008*), but we did not measure behavior *per se*, and so we cannot comment on exactly how food influenced nesting aside from nesting intervals. Timing of breeding as a response to food abundance is clearly important and can explain a decline in clutch size over time (*Murphy, 1986*). Also, high expected predation can reduce reproductive effort (*Zanette et al., 2003*; *Zanette et al., 2011*). However, in birds with potentially long breeding seasons and high predation risk, perhaps the best strategy is repeated nesting attempts (*Roper, 2005*). If so, then food abundance is especially likely to be important to allow repeated investment in eggs. Combining breeding season length with renesting intervals, we can see that ∼38 days in the 24 ha (small) fragment with 19–37 day renesting intervals, a maximum of two attempts can fit into a season. Indeed, only two attempts at most were initiated by pairs in that small fragment. With food supplementation in larger fragments, the combined benefits of increased clutch size, longer breeding season and more rapid renesting result in as many as five attempts (in the 112 ha fragment) and much greater potential reproductive success (in the 214 ha fragment). Thus, in the simulation, annual productivity may vary from 1.5 (control) to 3.1 (fed) fledglings pair$^{-1}$ yr$^{-1}$, with ≤30% of the population being unsuccessful in the 214 ha

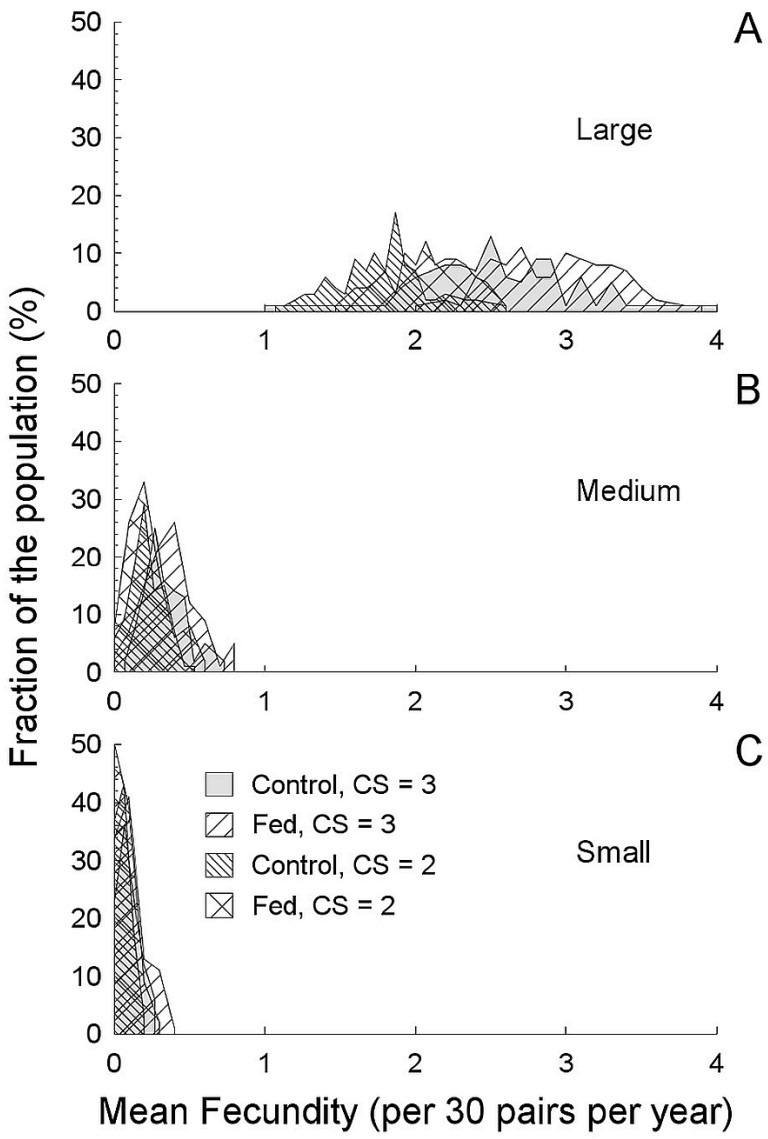

**Figure 4 Average annual productivity (fecundity) based on a simulated 30 pairs (a field population) and the breeding parameters described in this study (Table 3).** Average annual productivity (fecundity) based on a simulated 30 pairs (a field population) and the breeding parameters described in this study (Table 3), compared between fragment sizes and experimental food treatments (A, 214 ha. fragment; B, 112 ha fragment; C, 24 ha fragment). Increase fecundity in the largest fragment is a consequence of both reduced predation rates and food treatments. Note that the increased clutch size came at a cost of reduced time in which to breed, so while three-egg clutches produced more, the increase was not simply additive. The average productivity in both smaller fragments was less than one offspring per pair per year, and many "populations" (50% in the smallest fragment) were never successful.

fragment, in comparison with annual productivity always less than one fledgling pair$^{-1}$ yr$^{-1}$ and >90% unsuccessful in the 24 ha fragment and >75% unsuccessful in the 112 ha fragment (Table 3). In our simulation, when food was supplemented in the large fragment, all "populations" of 30 birds had an average predicted fecundity ≥1 (variable over the

range of 1 to 4), with the food supplmented treatments having the greatest values (>2.5). In stark contrast, in both smaller fragments, average predicted fecundity was always less than 1, regardless of supplemented food (even though when food is supplemented, success still increases, Fig. 4). The irregular curves in Fig. 4, rather than smooth lines, are due to the complex interaction between the number of days a nest survives, renesting intervals and breeding season length. For example, when nest success is greater, nests that fail last longer on average than at lower success rates, thereby reducing future opportunities to renest. This leads us to conclude that high predation rates can overwhelm the effects of food abundance.

Food abundance also influenced clutch size as has long been suggested (*Lack, 1947*; *Lack, 1948*; *Lack, 1949*; *Ricklefs, 1968*; *Ricklefs, 1980*; *Ricklefs, 2000a*; *Ricklefs, 2000b*; *Ricklefs, 2010*) and here, all but two of the 25 three-egg clutches (8%) followed food supplementation. Also, three-egg clutches were only found in the two largest fragments. Elsewhere at the same latitude, Variable Antshrikes can lay three-egg clutches and at least one has been found with four eggs (JJ Roper, pers. obs., 2003–2004). Thus, the more common two-egg clutch in this study may reflect local food limitation and also emphasizes the importance of the synergy between food and predation for nesting success.

The association between supplemental food and increased clutch size in the large fragments suggests complex associations between forest fragment size, food abundance and reproductive effort (*Donovan et al., 1995*; *Robinson et al., 1995*; *Burke & Nol, 1998*; *Vergara & Simonetti, 2004*; *Lloyd et al., 2006*; *Tieleman et al., 2008*). In some experimental studies, supplemental food resulted in no change in clutch size, which was then attributed to inflexibility in the genetic determination of clutch size (*Meijer & Drent, 1999*; *Bourgault, Perret & Lambrechts, 2009*). In our study, the impact of additional food was clear because supplementation began with the first nest attempt, and the increase in clutch size usually occurred after that first attempt when food was added. This result was different than expected for the Variable Antshrike because other species of *Thamnophilus* have a fixed clutch size of two (*Oniki, 1975*; *Skutch, 1985*; *Roper & Goldstein, 1997*; *Oniki & Willis, 1999*; *Roper, 2005*; *Roper, Sullivan & Ricklefs, 2010*).

Decreased nest predation rate in the largest fragment may be the result of several interactions (*Sinclair et al., 2005*; *Cain et al., 2006*; *Vergara & Hahn, 2009*; *Zanette et al., 2011*). Greater food abundance may have allowed reduced activity at nests (*Skutch, 1949*; *Skutch, 1985*) and greater nest attentiveness (*Chalfoun & Martin, 2007*). Greater distances to the edge in the large fragment may have reduced the likelihood of predators from the matrix between fragments from reaching the nests (*Gates & Gysel, 1978*; *Duca, Gonçalves & Marini, 2001*; *Sinclair et al., 2005*), although these fragments are natural and it is not clear whether the matrix has more or fewer predators than the forest itself. Alternatively, greater habitat heterogeneity within the larger fragments may reduce predator efficiency at finding nests by impeding the development of a "search image" by the predator (*Martin & Roper, 1988*). Perhaps simply increasing food abundance generates the perception that predation risk is lower, thereby allowing the increased clutch size (*Zanette et al., 2011*). Also, greater nest predation risk in smaller fragments may have created a

different landscape-of-fear which contributed to a shorter breeding season length and fewer attempts than the larger fragment (*Zanette et al., 2011*; *Bleicher, 2017*; *Creel, 2018*).

Here, with this food supplementation experiment in a tropical understory bird, we demonstrate a clear increase in reproductive effort with increased food availability. Evidence suggests that food abundance interacts with forest fragment size, even in natural forest fragments, resulting in reduced fecundity, reduced time in which to breed, and consequent loss of population viability, especially when predation risk is greater in the smaller fragments (*Porneluzi & Faaborg, 1999*). The combined effect of renesting faster, renesting more often and a longer breeding season results in the likelihood of nearly 100% of successful nesting every year. Thus, reproductive effort can increase with even a small addition of food in a fragmented forested landscape in southern Brazil, which suggests that annual reproduction may also vary widely among years, if food abundance also varies.

## ACKNOWLEDGEMENTS

We would like to thank the several field assistants that helped monitor nests and endured the long walks between forest fragments. Thanks to Vila Velha State Park for their help both in allowing the study in the park as well as providing housing and additional help. We thank anonymous reviewers for their constructive suggestions.

### Funding
James J. Roper was supported by a CNPq productivity fellowship (306963/2012-4). André M.X. Lima is supported by a doctoral fellowship (CAPES/REUNI). Angélica M.K. Uejima was supported by a CNPq doctoral fellowship. The funders had no role in study design, data collection and analysis, decision to publish, or preparation of the manuscript.

### Grant Disclosures
The following grant information was disclosed by the authors:
CNPq productivity fellowship: 306963/2012-4.
CAPES/REUNI.

### Competing Interests
James J. Roper is an Academic Editor for PeerJ. Otherwise, the authors declare there are no competing interests.

### Author Contributions
- James J. Roper conceived and designed the experiments, analyzed the data, contributed reagents/materials/analysis tools, prepared figures and/or tables, authored or reviewed drafts of the paper, approved the final draft.
- André M.X. Lima analyzed the data, contributed reagents/materials/analysis tools, prepared figures and/or tables, authored or reviewed drafts of the paper.
- Angélica M.K. Uejima conceived and designed the experiments, performed the experiments, analyzed the data, authored or reviewed drafts of the paper.

## Animal Ethics

The following information was supplied relating to ethical approvals (i.e., approving body and any reference numbers):

All methods followed standards of ethics under Brazilian law.

## Field Study Permissions

The following information was supplied relating to field study approvals (i.e., approving body and any reference numbers):

ICMBio/CEMAVE provided license for banding birds.

## Data Availability

The raw data and R script are provided in the Supplemental Files.

## Supplemental Information

Supplemental information for this article can be found online at http://dx.doi.org/10.7717/peerj.5898#supplemental-information.

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
