# Peer review of "Experimental food supplementation increases reproductive effort in the Variable Antshrike in subtropical Brazil"

_PeerJ, doi:10.7717/peerj.5898_

## Round 0.1 · original submission · Major Revisions

please revise according to the reviewers' comments. Pay particular attention to the reviewer who was more critical.

Reviewer 1 ·

Basic reporting

This is a nice experimental paper that will advance our understanding on tropical bird nesting strategies and fragmentation effects. In general I think this is a great paper is well written and experimentally test all hypothesis supporting some and providing new and interesting effect that fragmentation have on tropical bird nesting success.

Experimental design

Well done, however it was not clear if they also place the same containers where the mealworms where place, but empty, in the control territories.

Validity of the findings

The results and analysis are robust and appropriate for the experimental design and data.

Additional comments

In general I suggest expanding in some ideas in the discussion:

1) It is clear that food availability have an effect but is also clear that effect is stronger on smaller fragments less variation (at least in renesting intervals). I suggest that the authors should expand on this.

2) If you have data on female body mass it will be nice to know if female body mass differ between fragments, this may be also an important factor when considering reproductive effort.

3) It will be worth mentioning if the females that were successful and re-nested did it because the juveniles died or because the male took care of the nestlings.

4) Bi-parental incubation is common among antbirds (I suppose this species have bi-parental incubation), thus it will be relevant mentioning if male and female nesting effort change among fragments (if you have the data). I am just think that it could be possible that males help more in larger fragments than in small fragments due to higher natural food availability thus investment per nesting attempts will be lower, allowing individuals to expand breeding season, increase re-nesting attempts, etc. This could be also an alternative hypothesis that may explain your results. Males help more when there is more food.

5) I think is also important to discuss the possible causes of the variation observed in Figure 3 (Daily survival Rate in small fragments) and Figure 4 Renesting interval large fragment).

Specific comments

Page 11, Line 197: Do you have data on juvenile survival of the nest that were successful and females re-nested? If you don’t have detail data, I think it will be worth mentioned if you think if the juveniles died or the male took care of them by them self.

Page 13, line 245-247: This is a strong statement that I don’t think can be generalize for tropical species. The increment of clutch size with food abundance may be true for species were clutch size vary naturally (as you showed in this study), but there are studies with food supplementations that haven’t observed an increase in clutch size. Furthermore, I am not aware of a good example of clutch size increment with food supplementation in species that have fix clutch size (like most tropical bird species). If you do please cite those studies here and clarify, but if you don't I suggest avoiding making general statement here. Because your results cannot be generalize for most tropical birds that have a fix clutch size.

Page 13, Line 266-267: There are many paper that have conducted food experiments with interesting result that should be cited here, I suggest to expand on references search. I suggest to read and consider including Conway and Martin 2002, Sanz, J. J. 1996, Londoño et al. 2008 and Cucco & Malacarne (1997) as these references can add and expand to the current discussion on nest activity and food availability. Also all but (Conway and Martin 2002) conducted food manipulation and Skutch's papers cited here did not. Furthermore, some of the references that I suggest here showed that nesting bird with supplemented food do not always change the amount of trips to the nest. Thus, I think that studies that have not conduct food manipulation experiments will be helpful to make your point in this section.

Page 14, Lines 268-269: I suggest to site more recent papers See also Cox et al. 2012 The Auk, 129(1):147-155. 2012. Cox et al. 2012 Landscape Ecol 27:659–669.

Page 26: I suggest removing Figure 1.

Reviewer 2 ·

Basic reporting

No Comments

Experimental design

I found the research questions and predictions poorly described, especially with respect to the interactive effect of predation risk, fragment size and food availability on bird reproduction.
In addition, the sample sizes used in each treatment (control and fed-nests) are not clearly stated, so that I have troubles to evaluate the accuracy of the statistical analyses (i.e. with respect to fragment sizes).

Validity of the findings

Due to apparent very small sample sizes (i.e. number of nests) in the smallest forest fragment, and lack of spatial replication in fragment size categories (i.e. 3 study fragments: one of 24ha, one of 112ha and one of 214ha), I am sceptical about the actual possibility of testing the effect of fragment size and its interaction with the food treatment on bird reproductive parameters.

Additional comments

COMMENTS TO ROPER ET AL.
The present manuscript describes the effect of experimental food supplementation on some breeding components of a tropical passerine, the variable antshrike. The study has been conducted in three forest fragments of varying sizes, and the reproductive response of birds is compared among fragment sizes. I found the study interesting, especially because there are very few food supplementation experiments of that type at low latitudes and food allocation strategies in tropical birds still remain poorly understood. However, I found very surprising that this thread was not used at all in the manuscript (nothing about food allocation and reproductive strategies in tropical birds is discussed in the Introduction and Discussion (what are the predictions, how such strategies are predicted to differ compared to birds living in temperate areas?). In addition, the manuscript suffers from some serious drawbacks, including overall lack of structure (especially in the Introduction), wordiness, poor English syntax, unclear working hypotheses and statistical methods. This unfortunately renders the manuscript hard to follow. I list my comments below, in more details.
The Introduction suffers from a serious lack of cohesion, which creates some confusion about the main objectives of the study. Even the last part of this section, where the authors attempt to define their working hypotheses, is weak. The Introduction does not really match the expectations of the reader and the hypotheses that can be tested with the available dataset (but this feeling may be caused by the difficulty I had to follow the ideas and the logic of the statements). For example, throughout the whole Introduction, the authors attempt to explore varying scenarios involving interacting effects of food availability, predation risk and fragment size on varying components of bird reproduction, but the aim section only talks about testing the “importance of food abundance”: “we predict that food supplementation will result in increased reproductive effort, which may be manifest as increased number of nesting attempts, reduced intervals between nesting attempts, greater nesting success and combinations thereof”. No prediction is given for the effect of the two other factors and their interactions. I agree that these interactions are complex, but you need to make an effort to clarify the logic in your statements throughout the whole section. What are your expectations based on the literature, and most importantly, what can you actually test with your study design and the data in hand? This should be clearly summarised in the Aim section. I also recommend repeating concisely each question or hypothesis when you describe the statistical analyses, since you present several analyses, one for each of the four breeding responses recorded (number of nesting attempts, length of renesting interval, breeding season length, nesting success).
It is not clear what hypotheses you intend to test with respect to predation risk. Because you stress (legitimately) in the Introduction the interactive effect of forest fragment size and predation risk and present nest survival data in three fragments of varying sizes, my first impression was that you used fragment size as a surrogate of predation risk. Is that correct? If yes, do you assume that birds dwelling in small forest fragments face higher predation rates compared to larger fragments due to larger perimeter/area ratio (i.e. the boundary effect, which is by the way not mentioned at all), fewer nesting sites, less heterogeneity in the landscape structure and lower food availability? Nest survival data from control sites (presented in Figure 3) suggest it might be the case. Nothing is said on predator density or even predator identity in the different fragment sizes; is this data available?
To my opinion, the authors have missed some very important concepts around life-history and reproductive strategies in tropical birds. Resource allocation trade-offs of tropical birds are still poorly known. While it has been suggested that tropical resident birds should invest extra energy into self-maintenance rather than immediate reproduction (e.g. Ricklefs & Wileski 2002 Trends Ecol. Evol.), other studies have suggested that birds living at low latitudes are more plastic in advancing their timing of breeding, and therefore increasing their breeding success, than birds living at high latitudes, where factors other than food constrain life-history parameters (see e.g. Scheuerlein A, Gwinner 2002 J. Biol. Rhythms, and Ruffino et al. 2014 Front. Zool.). There are very few experimental studies testing the effect of food supplementation/limitation on the reproductive strategies of birds living in tropical areas, and to me this study has the potential to dig into this poorly known research area. The authors should devote a significant part of the Introduction/Discussion to these concepts.
Moreover, more should be said about the originality of the species. It is apparently a poorly known species and this should be stressed already in the Introduction. Why would you study this particular species? Is it of conservation concern?
As a consequence of the major points listed above, I strongly suggest rewriting most of the Introduction.
I found very hard to track sample sizes and there is nothing more frustrating to search for this information throughout the whole text. At the first mention, on line 97, it is only indicated that “about half of the pairs” were supplemented (how many exactly?). Then, on line 153-155, it is written that a total of 103 nests were found and monitored, but only three pairs were monitored in the smallest fragment, and eight pairs in the two others (which equals to 19 pairs in total, and not 103). Does the figure 103 nests include nests surveyed during 2001 and 2002)? Later, on lines 176-177, it is mentioned that “none of the seven nests were successful in the small fragment”, compared to five out of 63 in the medium-sized one, and 17 out of 29 in the largest one (which equals to 99 nests in total, and not 103).
It seems that the number of nests monitored in both the control and treated samples of the small fragment is very small (probably <10, maybe <5? Given the figures in Table 1, I tend to conclude that N = 1 for control nests and N = 2 for fed nests). If this is the case, I seriously doubt that these data would be useful in any of the analyses including Fragment Size as an explanatory variable. We can actually see from Figure 3 that the confidence intervals of mean nest survival rates in the small fragment are huge! In addition to very few forest fragments used in the analyses (N = 3) and a lack of spatial replication of each size category, I am very sceptical about the actual possibility of testing the interactive effect of fragment size and food availability on bird reproduction. If the sample size of the small fragment is too small, an alternative option is to drop those nests, merge the data from the two largest fragments and drop the fragment size hypothesis (since, anyway, you only have little power with the data in hand).
The statistical analyses are poorly described. I had a hard time understanding the structure of the models used. The information is patchy and it should be made clearer what variables are used in what models to answer what question.
On lines 211-212, it is written that the effect of fragment size on clutch size could not be tested due to low sample size. However, on lines 128-129, it is mentioned that clutch sizes were compared among fragment sizes with log-likelihood ratio tests. Did I miss something here?

Minor comments
Some examples where the structure of the text should be improved (i.e. lack of cohesion between sentences): Line 26-28: what is the link with predation risk mentioned in the previous sentence? Line 44-49: I don’t follow the logic between the two parts of the sentence. Lines 50-53: “When predation rates are high, renesting is even more important for annual reproductive success”: this sentence suggests that birds tend to renest more often to overcome high predation rates. However, the following sentence claims the opposite. Line 130-132: I really don’t understand this sentence.
Line 25: Ruffino et al. (2014-Front. Zool.) have recently published an updated meta-analysis on the effect of food supplementation on bird reproduction
Line 29-31: I don’t understand the rationale here (“because of its influence on both the number of nesting attempts and individual nest success”). “When breeding seasons are long”: are you referring here to multi-brooded species being more likely to respond to increase in food abundance, as was suggested by e.g. Svensson (1995-Anim. Behav.)?
Line 30: should correct to “…food limitation may affect annual…”
Line 50: should be corrected to “…between fragment size and both predation risk and food abundance…”
Line 74: Should the Latin species authority appear at the first mention of the Latin species name (i.e. in the Introduction)?
Line 85: should correct to “… size, are all very poorly known,…
Line 99: do you mean “domestic (animal?) cages”?
Line 100: what is the density of feeders provided by fragment (i.e. how many per active nest?). This information is lacking to evaluate whether or not the abundance of food was enough to trigger a reproductive response.
Line 104: Did you make any observations of intruders (from control areas or non-target species) feeding on the mealworms? This is important information in order to evaluate the efficiency of the food treatment.
Line 111-112: what do you mean by their interactions?
Line 112: should read “covariate”
Lines 114-119: unnecessary statements. If needed, the predictions can be rephrased in one short sentence.
Line 123: what do you mean by “The other variables are continuous”? Which ones? The ones that you cite are count variables (renesting intervals: number of days, number of renesting attempts).
Line 126-127: why not analysing also the effect of food on the renesting intervals of successful nests and comparing the results to those of failed nests?
Line 136: do you mean “covariates” by “correlated variables”? Were renesting intervals and breeding season length included in the annual productivity models?
Line 146-152: This part should be reduced and moved to the Methods (although I am not entirely convinced that it should be kept at all). This cannot be part of the results since there were no breeding birds and therefore no treatment. Same remark for the 2000 data that were not included in the analyses (lines169-173).
Line 180: 112ha instead of 105ha
Line 246-247: “all but two of the 25 3-egg clutches followed food supplementation”. Do you mean that only two of the 3-egg clutches observed were from non-supplemented areas?
Line 247: should read “two largest fragments”
Line 245-263: These two paragraphs are confusing and would need to be restructured with more logic between ideas and arguments. From the clutch size analyses, your results suggest that birds dwelling in the largest forest fragments are more food-limited (they increase their clutch size) than those from the small fragment (they don’t). However, you conclude the opposite on lines 249-251 or do you mean that food quantity was not enough in the small fragment to overcome a greater food stress or predation risk compared to the larger fragments?
Figures: please add sample sizes above each bar
Figure 6: the unit of the y-axis label should be “number of pairs.year-1”

---

## Round 0.2 · Minor Revisions

Thank you for your changes and the reasoned arguments behind the points you did not change. Your most critical reviewer has declined the opportunity to re-review, and my re-reading of the manuscript finds it largely acceptable. I would like you to fix a few small things:

Line 26 what is an income breeder
Lines 46-47 explain
Line 57 Manipulated
Line 99: why is "success" in here?
Line 112: why was the dish moved closer to the new nest
Lines 136-137 confusing. Should this be: while we predicted...because there is...?
Line 262: >1, widely variable...4, with the food supplemented...

---

## Round 0.3 · accepted · Accept

Thank you for making these final revisions!

#